# The Function of the Histamine H4 Receptor in Inflammatory and Inflammation-Associated Diseases of the Gut

**DOI:** 10.3390/ijms22116116

**Published:** 2021-06-06

**Authors:** Bastian Schirmer, Detlef Neumann

**Affiliations:** Institute of Pharmacology, Hannover Medical School, D-30625 Hannover, Germany; schirmer.bastian@mh-hannover.de

**Keywords:** inflammation, cancer, colitis, mast cell, epithelial cell, gut

## Abstract

Histamine is a pleiotropic mediator involved in a broad spectrum of (patho)-physiological processes, one of which is the regulation of inflammation. Compounds acting on three out of the four known histamine receptors are approved for clinical use. These approved compounds comprise histamine H1-receptor (H_1_R) antagonists, which are used to control allergic inflammation, antagonists at H_2_R, which therapeutically decrease gastric acid release, and an antagonist at H_3_R, which is indicated to treat narcolepsy. Ligands at H_4_R are still being tested pre-clinically and in clinical trials of inflammatory diseases, including rheumatoid arthritis, asthma, dermatitis, and psoriasis. These trials, however, documented only moderate beneficial effects of H_4_R ligands so far. Nevertheless, pre-clinically, H_4_R still is subject of ongoing research, analyzing various inflammatory, allergic, and autoimmune diseases. During inflammatory reactions in gut tissues, histamine concentrations rise in affected areas, indicating its possible biological effect. Indeed, in histamine-deficient mice experimentally induced inflammation of the gut is reduced in comparison to that in histamine-competent mice. However, antagonists at H_1_R, H_2_R, and H_3_R do not provide an effect on inflammation, supporting the idea that H_4_R is responsible for the histamine effects. In the present review, we discuss the involvement of histamine and H_4_R in inflammatory and inflammation-associated diseases of the gut.

## 1. Introduction

The biogenic amine histamine (2-(4-imidazolyl)-ethylamine) is a short-acting local mediator present in virtually all mammalian tissues. It is generated by the enzyme L-histidine decarboxylase (HDC) [1], and degraded either intracellularly by methylation (histamine N-methyltransferase; HNMT) [2] or extracellularly by oxidation (diamine oxidase; DAO) [3]. Cells commonly known to produce histamine are mast cells and basophils [4,5], intestinal enterochromaffine-like cells [6], and histaminergic neuronal cells [7]. These cells constitutively express HDC and produce histamine, which is bound to acidic polysaccharides such as heparin and stored in granules [8,9]. Release of histamine from these cells can take place by different mechanisms: piecemeal degranulation, kiss-and-run exocytosis, and conventional full fusion exocytosis [10,11]. Conventional exocytosis is induced by a specific stimulus, e.g., Fcε receptor-mediated activation of mast cells, and results in the rapid but transient occurrence of high local histamine concentrations. In contrast, piecemeal degranulation in mast cells is observed in diseased tissues and results in sustained presence of histamine, however, at lower concentration. Other cells, such as monocytes, dendritic cells, and T cells [12], produce significant amounts of histamine only during inflammatory reactions [13,14], which induce expression of HDC. Histamine produced in these cells is immediately released instead of being stored. In comparison to the release from storage granules, this leads to slower but sustained release kinetics and resulting tissue concentrations remain lower [15,16].

The biological functions of histamine are pleiotropic and comprise, but are not restricted to, the regulation of gastric acid release, neurotransmission, and pain perception/hypersensitization as well as the contribution to inflammation. Due to the widely used anti-histamines, the most commonly known role of histamine in inflammation is to promote allergic reactions. But beyond that, histamine is involved in manifold key aspects of inflammation, such as shaping the Th1/Th17/Th2-balance and regulating access of other mediators and inflammatory cells to inflamed tissue by acting on resident immune cells and vascular endothelial cells. Moreover, pain and itch sensation from inflamed tissue are modulated by histamine acting on neuronal cells. The pleiotropic effects of histamine on inflammatory processes have been reviewed in detail elsewhere [17]. These different biological functions of histamine rely on the existence of four histamine receptor subtypes and their tissue and cellular distribution [18]. All the histamine receptor subtypes belong to the class A rhodopsin-like G protein-coupled receptors (GPCRs) and have been named histamine H1-receptor (H_1_R), H_2_R, H_3_R, and H_4_R in order of their discovery. While H_1_R and H_2_R are ubiquitously expressed and bind histamine with relatively low potency (pEC50 ~6), the expression of high potency (pEC50 ~8) histamine receptor subtypes H_3_R and H_4_R is rather restricted [19]. While H_3_R is found on mammalian neuronal cells [20], H_4_R originally has been identified in immune cells, such as mucosa-associated immune cells of the intestine. A more detailed summary of immune cells expressing H_4_R is presented in Table 1 (Table 1); note that caution has to be paid, since not all of the data provided in the cited literature have been verified by two independent methods (see discussion below). Meanwhile also non-immune cells, such as epithelial cells of the gut, have been shown to express H_4_R [21]. Thus, due to its expression pattern, H_4_R is an attractive pharmacological target to treat e.g., inflammatory reactions of the gut [22,23,24,25].

## 2. H_4_R Basics

The existence of a distinct histamine receptor mediating calcium mobilization in eosinophils has been claimed already in 1994 [80]. Discovery and cloning of the corresponding H_4_R were published in the year 2000 quasi-contemporaneously by several independent groups. In essence, H_4_R was identified based on its relative high genetic homology (37%) with H_3_R, while showing less than 30% homology with H_1_R, H_2_R, and other biogenic amine receptors [26,60,70,81,82,83]. Engagement of agonists entails recruitment of Gαi proteins to H_4_R that subsequently leads to activation of phospholipase C as well as to inhibition of membrane-bound adenylyl cyclase activity. As a consequence, calcium ([Ca^2+^]_i_) is mobilized from intracellular stores and the concentration of cytosolic cAMP is diminished. Independent of G protein-signaling, β-arrestin 2 is recruited to agonist-bound H_4_R, initiating mitogen-activated protein kinase (MAPK) cascade activation as well as receptor desensitization and internalization [84,85,86,87]. Activation of the H_4_R signaling pathways lead to induction of pro-inflammatory AP-1 in Th2 cells and monocyte-derived dendritic cells (MoDC) [27]. Furthermore, production of the Th1-associated cytokines IL-12 and IP10 in MoDC is reduced by engaging the H_4_R, indicating that histamine might shape a Th2-biased immune response via these mechanisms [36,37,88]. Attraction and activation of eosinophils, mast cells, and dendritic cells could also be shown to be mediated by H_4_R activation, thus further amplifying the Th2 inflammatory phenotype [37,38,49]. If activation of H_4_R-signalling can be exploited to treat Th1/Th17-driven pathologies such as Crohn’s disease by inducing a Th2-bias remains controversial [89,90].

Pharmacological ligands, either agonists or antagonists/inverse agonists, or in vivo models using genetically modified animals, are the main tools to identify H_4_R functions on specific cells or tissues in health and disease. Compound JNJ7777120 (1-[(5-Chloro-1H-indol-2-yl)carbonyl]-4-methylpiperazine) was the first selective H_4_R antagonist developed and it has been used in a large number of laboratories in in vivo and in vitro analyses [91]. JNJ7777120 has turned out a biased ligand at the human H_4_R, inversely agonizing G protein-dependent signaling, while agonizing G protein-independent signaling. Thus, the results gained so far must be re-evaluated accordingly [92,93]. Moreover, it was claimed that JNJ7777120, while acting as inverse agonist at the human H_4_R, is an agonist at the mouse and rat H_4_R. These results, however, were obtained using membrane preparations of insect cells expressing recombinant human H_4_R together with selected G proteins and have never been reproduced in a native setting [94]. Unfortunately, a comprehensive evaluation of the activities of JNJ7777120 in different species remains to be performed. Alternative to JNJ7777120 other ligands at the H_4_R are available, providing agonistic and inverse agonistic/antagonistic activities. However, their use may be afflicted with problems concerning specificity, effectivity, and potency, too. Thus, in a complex biological system, studies on receptor function should not be based on a single selected ligand. At least chemical different ligands have to be employed or complementary experimental approaches must be applied. For example, genetic manipulation of the receptor to be studied can be used in addition to ligand-driven experimental systems. Of course, also vice versa, data generated by a genetic-based approach alone should be very carefully validated unless proved by an independent experimental system.

Mice are widely used to model human diseases since their handling is relatively easy and inexpensive. Furthermore, a huge array of different strains with defined genotypes is available. Among these are mice with a targeted mutation in the Hhr4 locus, which encodes the mouse H_4_R. These mice do not express a functional H_4_R and without any manipulation they do not demonstrate any biological abnormalities as compared to their wild type (WT) counterparts [49]. However, upon experimental induction of allergic diseases such as ovalbumin-induced asthma or experimental dermatitis, the lack of functional H_4_R leads to reduced symptoms [39,40,95]. These results have been reproduced by applying pharmacologic blockade of the H_4_R, and thus, the involvement of H_4_R in experimental asthma and dermatitis in mice is widely accepted. Unfortunately, in humans H_4_R antagonists did not gain the anticipated results regarding effectiveness and safety [96] (clinical trials.gov: NCT01823016 (asthma), NCT01493882 (asthma), NCT03517566 (dermatitis)). This discrepancy may be based on the H_4_R expression pattern, which is similar, but not necessarily identical in mice and man, as exemplarily demonstrated in Table 1 for immune cell types (Table 1). Whether this holds true for other tissues and cell types such as the gut as well remains to be analyzed in detail. Thus, if mice are able to serve as model for human H_4_R pathophysiology, has to be evaluated in a context (i.e., tissue/cell type)-dependent way. Using different approaches to manipulate H_4_R function, it became evident that of all different immune cells, H_4_R mainly regulates human and mouse mast cells and eosinophils, e.g., their migration and degranulation [38,49,50,71,72,97,98]. Because mast cells are the main producers of histamine and at the same time respond to histamine via the H_4_R, it is intriguing to speculate about the biological meaning of this auto-regulatory cycle. In eosinophils, the H_4_R seems to be only a minor player in comparison to classical activators and may be responsible for the cellular activities’ fine tuning [73]. For the evaluation of histamine effects, it also has to be taken into account that most cells do not exclusively express H_4_R. The observed effects elicited by histamine in cells, tissues, or animals are always the sum of effects elicited by all histamine receptor subtypes present, which are H_1_R, H_2_R, and H_4_R in the case of eosinophils and mast cells. This scenario becomes even more complicated, since, as described above, the histamine receptor subtypes possess different affinities for histamine and effective interstitial concentrations of histamine cannot be reliably measured so far.

## 3. Histamine in the Intestine

Inflammatory bowel diseases (IBD) are idiopathic, chronic-recurring diseases of the gut. Their two main manifestations, ulcerative colitis (UC) and Crohn’s disease (CD), differ in their clinical, endoscopic, and histologic appearance. In CD, the inflammation appears in diffuse lesions that can be found all over the digestive tract and deeply penetrates the intestinal wall, possibly affecting all layers. In contrast, inflammatory lesions in UC start in the rectum, proceed upwards but do not exceed the colon, and remain superficial at the mucosa [99,100,101,102]. CD and UC also differ regarding the immune response: while CD is largely associated with a Th1/Th17-dominated response, UC is mostly defined by a Th2-response [103,104,105]. Nevertheless, both ailments evoke a series of similar symptoms (e.g., mucosal lesions, ulcera, edema, diarrhea, bloody stool, abdominal pain) severely affecting the quality of patients’ lives and eventually limiting their life expectancy through extra- and intra-intestinal complications such as colorectal cancer (CRC) [100,101,106]. The risk to develop such complications is further augmented by the currently applied treatment schemes, which are based on immunosuppressive drugs such as 5-aminosalicylic acid or glucocorticoid receptor agonists. These drug demonstrate remission rates of only 50%, but long-term treatment can induce and/or support immunosuppression-related disorders [100,101,107].

It has been known for quite a long time that in affected colorectal mucosal samples obtained from CD and UC patients histamine concentrations are elevated in comparison to samples from unaffected sections or healthy control persons [108,109]. A first experimental hint that histamine may be involved in inflammatory diseases of the gut was identified in a model of acute colon inflammation, i.e., dextran sulfate sodium (DSS)-induced colitis. Comparing wild type mice to mice deficient in HDC expression, lack of HDC-mediated histamine synthesis resulted in reduced clinical symptoms, while inflammation of the colon was not affected. The authors attributed these findings to the reduced number of IL-10-producing lymphocytes in the colonic mucosa, probably reducing, but not abolishing, a Th2 response, and the altered composition of fecal bacteria [110]. Unfortunately, the explanation why reduced IL-10 production, commonly regarded as an anti-inflammatory cytokine, leads to reduced clinical symptoms but does not affect colonic inflammation was somewhat sparse. In fact, the possibility that IL-10 evokes differential functions in seemingly identical settings is quite well described. While mice devoid of IL-10 production commonly serve as models that develop severe enterocolitis, the susceptibility to develop colitis largely depends on the complex gut microbiota [111,112]. Thus, for the study performed by Bene et al. [110], while they excluded the contamination of their system by alimentary-provided histamine due to feeding a histamine-free diet, it would have been advantageous to analyze the microbiota of the mice, or, at least, the presence of histamine-producing species such as *E. coli, M. morganii*, or *L. vaginalis* [113].

## 4. Histamine Receptors in the Intestine

The detection of functional expression of H_4_R as well as of other GPCR is a still ongoing discussion that currently lacks impetus. Since in 2012 the specificity of H_4_R-selective antibodies was questioned [114,115], a comprehensive processing of this issue has not taken place. Thus, antibodies recognizing H_4_R that have been rigorously evaluated according to some general rules [116] are still missing (or at least the data demonstrating the rigorous evaluation have not been provided). These kinds of problems are not specific for the detection of H_4_R proteins by selective antibodies. Other methods such as mRNA detection by RT-(q)PCR or functional assessment using agonistic or antagonistic ligands also bear uncertainties. RT-PCR is a very sensitive detection method, but, besides the risk that trace amounts of contaminating genomic DNA may pose the presence of specific mRNA [117], it is not known which degree of mRNA expression would enable effective receptor expression [118]. The use of receptor ligands is afflicted with problems similar to those of the antibodies; selectivity, which is strictly concentration-dependent, is the major concern [18,119]. Thus, eventually at least two independent techniques should be provided in order to reliable demonstrate functional H_4_R expression. Alternatively, genetic approaches (knock out/knock down cells, tissues, or animals) are to be used to support expression data gained by other methods. In particular, targeted knock-out approaches would be able to foster our knowledge on the cell type-specific expression of H_4_R, being applicable not only to the intestine but to all other tissues as well.

Since histamine seemed to be functionally involved in gut (patho)physiology, studies aimed at comprehensively identifying the histamine receptor subtypes expressed in the intestinal tract using tissues derived from different species [24,120,121,122,123]. Summarizing these and other studies, all histamine receptors except H_3_R are expressed throughout the intestinal tract from mice, rats, monkeys, and humans (Table 2), while the data gained in dogs are questionable due to the use of not fully reliable methods [120]. From a quantitative point of view, it also appeared that H_4_R expression is significantly less abundant in comparison to H_1_R and H_2_R, at least on the mRNA level [24,121]. Subsequently, a possible function of H_4_R was investigated using pharmacological and genetic approaches. In rats suffering from trinitrobenzene sulfonic acid (TNBS)-induced colitis, the binding of H_4_R by JNJ7777120 resulted in reduced inflammatory symptoms [124], indicating indeed a pathological involvement of H_4_R in the colon. Following studies using mice either or not deficient in H_4_R expression and using JNJ7777120 in models of chemically-induced colitis confirmed this indication [89,90]. The data discussed so far strongly indicate that the local mediator histamine is present in high concentrations at inflamed sites of gut inflammatory diseases and that histamine affects the inflammatory pathology. However, they do not provide any evidence on the cell types involved.

## 5. Histamine-Responsive Cells

As discussed above, mast cells, allocated to the innate immune system, do not only produce histamine, but also respond to stimulation of the H_4_R. Besides the contribution of mast cells to allergic inflammation [132], they also seem to be involved in non-allergic inflammatory diseases of the gut, since in the intestine of IBD patients, parallel to mucosal histamine concentrations, number and activity of mast cells are increased [133,134]. Moreover, few data indicate that the application of cromolyn, a mast cell-stabilizing drug, affects experimentally-induced pathogenesis of gastric cancer and ischemia/reperfusion injury of the ileum [135,136]. In the pathogenesis of experimental colitis in mice, however, mast cells seem to be dispensable [137,138]. Thus, histamine apparently is generated also by sources other than mast cells (Figure 1). Indeed, using mast cell-deficient rats, it was demonstrated that the absence of intestinal mast cells reduced mucosal histamine concentrations, but did not nullify them [139]. Expression of the histamine-generating enzyme HDC can be induced in myeloid cells and probably also in virtually every other cell type [140,141]. As a possible alternative source of histamine in mouse intestinal pathology, CD11b^+^Ly6G^+^ immature myeloid cells have been described, since they are involved in inflammation-associated carcinogenesis in the gut [142]. Whether or not a human counterpart does exist, at least a functional one, has not been elaborated so far. Moreover, although originally identified in another model, dendritic cells are thought to be able to generate histamine [39,40,98,143]. Thus, dendritic cell-generated histamine may contribute to intestinal inflammation, too. This, although initial data argue against it [23], remains to be elucidated.

Besides mast cells, enhanced numbers of eosinophils have been identified in the inflamed gut mucosa of IBD patients, too [144,145]. The role of these innate immune cells in the pathogenesis of colitis, however, is still controversially discussed, either promoting pathogenesis or limiting inflammation [145,146]. Eosinophils are generated in the bone marrow, released into blood circulation, and eventually migrate to target tissues with the gut being a main destination. Eosinophil migration is mainly regulated by the chemokine eotaxin-1. As already mentioned above, it has been shown in vitro that histamine affects this process, albeit to a lesser extent, by its interaction with H_4_R on eosinophils [38,71,72,73,80]. Thus, one may speculate that this process is active in the inflamed colon, facilitating histamine to add to the attraction of eosinophils (Figure 1). Consequently, this hypothesis favors a pro-inflammatory function of eosinophils, since blockade of H_4_R activity, which diminishes experimental colitis, would also reduce number and, thus, activity of eosinophils in the affected intestinal areas. [21,147,148].

Neutrophils belong to those cells first recruited to the site of inflammation, where they combat the potentially harmful intruders applying rather unspecific mechanisms, condoning collateral damage at healthy cells and tissues. Consequently, neutrophils can be detected in high numbers in the colitic mucosa, probably dependent on the enhanced expression of IL-6 and downstream CXCL-chemokines (Figure 1), mediators that activate/attract these cells [149,150]. In H_4_R knockout mice as compared to WT mice, reduced numbers of colon-infiltrating neutrophils as well as lower colon tissue concentrations of IL-6, CXCL1, and CXCL2 have been detected upon induction of experimental colitis [23,89]. Thus, it is most probable that histamine via H_4_R increases neutrophil infiltration during colitis, presumably in an indirect manner [151,152]. However, the precise cellular sources of histamine/H_4_R-regulated production of IL-6, CXCL1, and CXCL2 remain ill defined. The related chemokines CXCL1 and CXCL2, which both bind to the receptor CXCR2, are synthesized by variety of immune cells including macrophages and neutrophils themselves, and by epithelial cells. Macrophages have been shown to respond to histamine via H_4_R [64,65], however, not in the context of intestinal diseases so far. Whether or not neutrophils functionally express H_4_R is still matter of debate. Eventually, epithelial cells are the most probable source of CXCL1 and CXCL2 in the context of colonic inflammation.

Colonic epithelial cells comprise a group of cells forming a tightly linked monolayer epithelium lining the inner surface of the intestine and providing, by several means, a barrier between the gut lumen and the host tissue. Impairment of this barrier would result in the luminal content gaining access to intra- or sub-epithelial areas, followed by an inflammatory reaction [153,154]. New intestinal epithelial cells are generated throughout the life of the host by proliferation of intestinal epithelial stem cells residing in the base of each intestinal crypt. Due to the continuous proliferation of these stem cells, the resulting daughter cells move forward towards the tip of the crypt or villus, where they eventually are scaled off and die. During this movement, the cells differentiate into specialized lineages, providing the different functions of the intestinal epithelium. The majority of developing cells become absorptive enterocytes, responsible for selective uptake, metabolism, and transport of nutrients. Other cells differentiate into mucus-secreting goblet cells that produce two layers of mucus with different densities at the luminal surface, adding to the colonic barrier function. The integrity of the cellular barrier is provided by close intercellular junctions created by tight junction proteins, adherens junction proteins, and desmosomes, omitting epithelial transfer by diffusion while enabling regulated uptake by either trans- or para-cellular pathways (reviewed in [155,156]). There are some indications that colon epithelial cells of mouse and men express H_4_R [89], although others conclude controversially [23]. Nevertheless, using organoid-derived mouse colonic epithelial cells, a direct in vitro effect of histamine on the trans-epithelial electric resistance is detected. Since the employed cell culture setting is devoid of any other cell type than epithelial cells, a direct effect of histamine on the epithelial cells could be concluded (Figure 1). Finally, this effect was inhibited by either the application of JNJ7777120 or by the use of cells collected from organoids formed of H_4_R-deficient colon epithelial cells, both pointing to a direct effect of histamine at H_4_R expressed on mouse colon epithelial cells [89]. Moreover, human colon epithelial cell-derived organoids that were micro-injected with FITC-dextran and either or not stimulated with histamine indicate that histamine promotes leakage of the dyed macromolecule out of the organoid’s lumen (unpublished).

In summary, it seems probable that histamine via H_4_R contributes to colonic inflammation, mostly to the innate arm by regulating production and release of the pro-inflammatory mediators TNF, IL-6, CXCL1, and CXCL2 promoting neutrophil recruitment. In addition, in vitro the epithelial cell barrier is battered by histamine/H_4_R as well, possibly contributing to inflammation. Whether or not colon epithelial cells also are responsible for the histamine/H_4_R-triggered enhanced release of the mediators mentioned above is not definitively settled so far.

## 6. Histamine and Carcinogenesis

Cancer, in general, is a major public health problem and does not only account for a high physical strain of the patients but also for intensive emotional pressure in both patients and their relatives and, finally, for a large number of deaths. Although anti-cancer therapies, including new strategies and compounds, have advanced strongly in the past decades, they are still loaded with detrimental effects, such as toxicity or poor response [157]. These effects are, at least in part, due to the high pathophysiological and genetic heterogeneity of the diverse forms of cancer, rendering it necessary to develop new therapeutic options that meet the different requirements of different cancers. A common feature of all forms of cancer is the inadequate cellular proliferation, resulting in the tumor. Thus, anti-cancer drugs aim at inhibiting proliferation and/or inducing death of cancer cells, and a major task for modern therapeutics is to provide these functions in a cell type- and probably differentiation status-specific manner [158]. A tumor cell can be defined by the presence of a specific pattern of molecules, which may represent targets for a specific tumor therapy. A single cellular receptor hardly defines a specific cell type/differentiation status, but may create a group of cells that reasonably can be targeted, avoiding severe adverse effects occurring when using broad acting anti-neoplastic drugs/therapies such as 5-fluoruracil, hydroxycarbamide, or ionizing radiation. Moreover, cellular receptors may be used as targets for adjunctive drugs, able to support the main drug and/or to reduce its effective dose, and thereby in summary reducing the risk of adverse effects, as exemplified by the H_2_R antagonist cimetidine in colorectal cancer patients [159,160]. In this respect, the endogenous agonist histamine in combination with IL-2 is approved to treat patients in the post-chemotherapy phase of acute myeloid leukemia [161], in which it reduces the risk of relapse. Although neither the molecular mechanism nor the histamine receptor subtype(s) involved have been determined so far, these clinical data place histamine in the list of possible anti-cancer drugs [162]. Thus, histaminergic drugs may be used in other types of cancers and substances selectively addressing specific histamine receptor subtypes as agonists or antagonists may provide therapeutic advantages [163,164].

CRC is the third most common type of cancer in humans and accounts for high mortality. UC patients are at a 2- to 3-fold higher risk to develop CRC in comparison to individuals not suffering from IBD and severity, extent, and persistence of inflammation positively correlates to carcinogenesis. In addition to colitis-associated CRC, CRC appears also sporadic and hereditary [165,166]. Although in these cases inflammation rarely precedes CRC, anti-inflammatory drugs are effective in preventing or delaying the disease [167,168]. Thus, inflammatory reactions seem to be involved in tumorigenesis of non-colitis-associated CRC, too. Consequently, it is tempting to speculate that histamine, in parallel to colon inflammation, promotes CRC, and, thus, that interventions blocking histamine activity reduce carcinogenesis. Consistently, in the colonic mucosa of CRC patients, HDC activity as well as histamine content are increased in comparison to normal samples [125,169]. However, in mice with experimentally induced CRC, deletion of HDC resulted in enhanced tumorigenesis as compared to wild type mice, pointing towards an anti-carcinogenic effect of histamine, which is supported by the finding that gut microbiota-derived histamine suppresses colorectal tumorigenesis [170]. The multiple effects histamine may exert in CRC, at least in animal models, can explain this contradiction. On the one hand, histamine promotes the underlying inflammatory process, leading to tumor initiation. On the other hand, histamine in the tumor’s tissue may affect differentiation of immature myeloid cells towards neutrophils and myeloid-derived suppressor cells, both resulting in tumor regression (Figure 1) [142,171]. While the effect of histamine on the differentiation towards neutrophils is a direct one, the differentiation of myeloid-derived suppressor cells is affected by IL-17, which is produced by tumor-associated mast cells upon histamine stimulation. This anti-cancer effect of histamine is supported by similar findings obtained in models of esophageal squamous carcinoma [172]. Interestingly, mast cells have been found to be abundant in colon carcinoma and to promote carcinogenesis in chemically-induced CRC in mice [173], and are associated with a poor prognosis in human CRC patients [174]. Others, however, reported that blocking histamine receptor function in a mouse model of CRC ameliorates disease symptoms [175]. These differences may be due to the different models used, concerning the experimental schedule and the species. Regarding the modes of action of histamine discussed above, the question arises if specific receptor subtypes are responsible for these different effects of histamine on inflammatory cells and tumor-regulating cells. In analogy to the pro-inflammatory effect of histamine via the H_4_R that has already been discussed above, the absence of H_4_R expression also leads to a reduction of chemically-induced carcinogenesis in mice [131]. Whether this is due to the reduced inflammation promoting carcinogenesis or to an effect of histamine on the tumor cell itself is unknown at present. However, some indications arise from the observation that the expression of H_4_R decreases in gastric carcinoma during progression, accompanied by the attenuated histamine-induced suppression of proliferation [172,176]. This idea is supported by our own unpublished data indicating that human colon cancer cell lines, which mostly arise from late stage carcinomas, do not functionally express H_4_R. It is worth noting that others have found H_4_R expression in such cell lines [125], which may reflect differences in details of the detection systems or the development of laboratory-specific cellular sub-lines. In summary, these data strongly indicate that the function of histamine in general and of the H_4_R in detail is far from being finally settled.

## 7. Conclusions

Histamine is a mediator that is mainly recognized due to its function in allergy, but takes part also in non-allergic inflammatory reactions. In the gut, the involvement of histamine and H_4_R in IBD, especially in the colon, has been demonstrated in different model systems by several laboratories, who mainly conclude a pro-inflammatory function (Figure 1). Cellular and molecular details of the H_4_R function in colon inflammation, however, remain elusive. In addition, tissues with minor indications on histamine/H_4_R responsiveness (smooth muscles, neurons) have not been discussed in this review. Even less differentiated is the picture on the effect of H_4_R in colon carcinogenesis. It is likely that H_4_R due to its pro-inflammatory function promotes the pathogenicity of colitis-associated CRC, but an additional effect of H_4_R on tumor or accessory cells currently can neither be excluded nor confirmed. Therefore, still a lot of work on histamine function on colon carcinogenesis has to be envisaged.

## Figures and Tables

**Figure 1 ijms-22-06116-f001:**
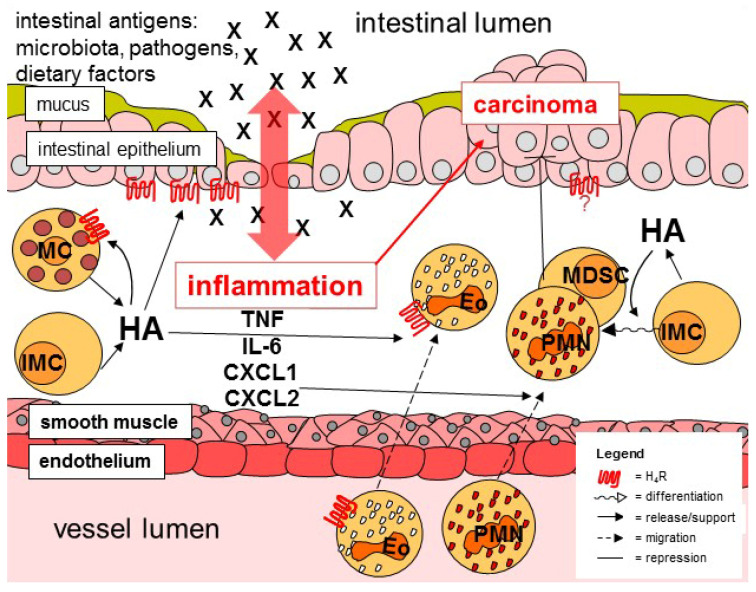
Schematic representation of inflammatory and carcinogenic mechanisms in the colon. MC, mast cell; IMC, immature myeloid cell; HA, histamine; Eo, eosinophil; PMN, neutrophil; MDSC, myeloid-derived suppressor cells.

**Table 1 ijms-22-06116-t001:** Expression of H_4_R in immune cells (mRNA and/or functional testing).

Cell Type	Mouse	Human	Reference
CD4^+^ T cell		x	[26]
Th2		x	[27]
Th9		x	[28]
Th17	x	x	[29,30,31]
Treg	x	x	[32,33]
γδ T cell		x	[34]
CD8^+^ T cell		x	[35]
DC	x	x	[36,37,38,39,40,41,42,43,44,45,46,47]
NKT	x		[48]
Mast cell	x	x	[26,49,50,51,52,53,54,55,56,57,58,59]
Neutrophil	x	x	[23,26,60,61]
Monocyte		x	[26,36,60,62,63]
MΦ	x		[64]
M1		x	[65,66]
M2		x	[67]
Basophil		x	[49,68,69]
Eosinophil	x	x	[26,38,49,60,70,71,72,73,74,75,76,77]
NK cell		x	[43,78,79]

**Table 2 ijms-22-06116-t002:** Studies on H_4_R in the gastrointestinal tract.

Main Findings	Species	Ref.
H_2_R and H_4_R are pro-proliferative and pro-angiogenic in HT29, Caco-2, and HCT116 colon cancer cell lines	human	[125]
H_4_R possesses pro-inflammatory role in TNBS-induced colitis in rats	rat	[124]
H_1_R, H_2_R, H_4_R are expressed in the human gastrointestinal tract; expression is altered in patients with gastrointestinal diseases	human	[24]
H_1_R, H_2_R, H_3_R, H_4_R activation excite human enteric neurons	human	[25]
H_1_R, H_2_R, and H_4_R are expressed in colon carcinoma and in adjacent normal mucosa; H_1_R and H_4_R expression is reduced in carcinoma compared to normal colon	human	[126]
H_1_R, H_2_R, and H_4_R are expressed in simian colon smooth muscle cells	mouse, monkey	[121]
H_4_R activity contributes to radiation-induced cytotoxic and genotoxic damages in small intestine	rat	[127]
H_4_R stimulates the release of DAO, contributing to histamine deamination during fat absorption	rat	[128]
H_4_R and H_1_R contribute to post-inflammatory visceral hypersensitivity	rat	[129]
H_4_R possesses a pro-inflammatory role in DSS-induced colitis in mice	mouse	[89]
H_1_R and H_4_R regulate the DC-CD4+ T-cell axis in peanut-induced intestinal allergic responses	mouse	[130]
H_4_R possesses an anti-inflammatory role in TNBS-induced colitis in mice	mouse	[90]
H_4_R possesses a pro-inflammatory role in experimental colitis in mice	mouse	[23]
H_4_R is functionally expressed on colon epithelial cells, affecting epithelial barrier integrity	mouse	[22]
H_4_R is involved in carcinogenesis of chemically-induced colitis-associated colorectal carcinoma	mouse	[131]

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
