# Peer review of "The Function of the Histamine H4 Receptor in Inflammatory and Inflammation-Associated Diseases of the Gut"

_ijms, 2021, doi:10.3390/ijms22116116_

Round 1

Reviewer 1 Report

Schirmer and Neumann herein present a review article on the involvement of histamine and one of their specific receptor, i.e., H4R in inflammatory and inflammation-associated diseases of the gut. This is a well-conceptualized and summary of findings from both rodent and human studies. The article is further supported only by one figure, presenting a scheme of inflammatory and carcinogenic mechanisms in the colon. This work presents a helpful and novel contribution to the histamine research field. However, in my opinion, there are several major points that remain to be accurately addressed. I believe that Reviewer suggestions are important for improving this paper. Without these corrections, the paper cannot be published. So I recommend a major revision.

  1. The basic information regarding histamine and H4R is very useful in the understanding of the conceptualization of this paper. However, some important aspects are missing. Namely, the authors stated that „release of histamine from cells takes place upon a specific stimulus, e.g. Fce-receptor-mediated activation of mast cells, and results in the rapid, but transient occurrence of high local histamine concentrations”. But it should be kept in mind that mast cells can release their granule contents but retain the granule membranes within the cytoplasm in the process called piecemeal degranulation. Thus, mast cells constantly secrete e.g. histamine which affects adjacent cells and tissues. This should be mentioned and briefly discussed in relation to the topic of the presented paper.
  2. The information regarding the presence of H4R in immune cells should be precise. Which immune cells express this receptor? Authors may present this data in the table.
  3. The role of histamine in the inflammatory processes, including the effects on other cells should be described in detail. Also, inflammatory consequences of initiating the signaling pathways associated with H4R activation have to be provided.
  4. It is known whether and how using mast cell stabilizers, e.g., cromoglicic acid affect inflammatory and inflammation-associated diseases of the gut? Are available any studies in this field?
  5. The table summarizing available studies (human or rodent) regarding histamine in inflammation-associated diseases of the gut would be very useful.

Author Response

Dear editor,

Thank you very much for handling our submission and considering its publication in IJMS. We also sincerely would like to thank the reviewers for their thoughtful and constructive comments, which helped to substantially improve our manuscript. In the following, you will find a point-by-point explanation of the revisions we made according to the reviewers comments.

Reviewer 1

1. The basic information regarding histamine and H4R is very useful in the understanding of the conceptualization of this paper. However, some important aspects are missing. Namely, the authors stated that „release of histamine from cells takes place upon a specific stimulus, e.g. Fce-receptor-mediated activation of mast cells, and results in the rapid, but transient occurrence of high local histamine concentrations”. But it should be kept in mind that mast cells can release their granule contents but retain the granule membranes within the cytoplasm in the process called piecemeal degranulation. Thus, mast cells constantly secrete e.g. histamine which affects adjacent cells and tissues. This should be mentioned and briefly discussed in relation to the topic of the presented paper.

Thanks for your kind evaluation and for the important suggestion. We introduced the process of piecemeal degranulation in the Introduction section.

2. The information regarding the presence of H4R in immune cells should be precise. Which immune cells express this receptor? Authors may present this data in the table.

We comprehensively compiled such data from the literature and summarized them in table 1 in the revised version of the manuscript.

3. The role of histamine in the inflammatory processes, including the effects on other cells should be described in detail. Also, inflammatory consequences of initiating the signaling pathways associated with H4R activation have to be provided.

Entirely answering this point, would gain material for a separate manuscript, thus it would blow up the present submission. Therefore, we decided to only briefly address this issue and added respective paragraphs to the Introduction and H4R basics sections.

4. It is known whether and how using mast cell stabilizers, e.g., cromoglicic acid affect inflammatory and inflammation-associated diseases of the gut? Are available any studies in this field?

Indeed, some few studies analyzing cromolyn in intestinal inflammatory diseases are available. We added these in the Histamine-responsive cells section.

5. The table summarizing available studies (human or rodent) regarding histamine in inflammation-associated diseases of the gut would be very useful.

This table has been generated and added to the manuscript as table 2.

Reviewer 2 Report

In this review manuscript the authors have detailed functions of histamine, binding of its receptors, particularly H4R, with an emphasis on pathophysiological role in the gastrointestinal tract. It is a quite good summary of the current knowledge of the topic. The language could be slightly improved, and although already well references, there are sections without any. Find enclosed a list of a few point.

Add a box with key points from the review.

Line 37. Mention other cells that produce histamine (Other cells, such as..)

Line 57 add “due to its expression pattern” to “Thus, [due to..] H4R is..”

Line 88-91. Rephrase the sentences “At least chemical..” and “This requirement...” to improve the language.

Line 92: Is the H4R expression pattern different comparing humans and mice?

On line 103 is detailed trials to block H4R. Are these trials for asthma and dermatitis as mentioned on line 101? In the abstract is mentioned trials for rheumatoid arthritis and psoriasis. Could the authors briefly add information of these.

Line 107-108. Is H4R expressed on both human and mice eosinophils and mast cells?

Line 171. Related to genetic approaches to study H4R, particularly for studies in the gut (inflammation and carcinogenesis, eg. line 311-327): targeted approach to remove the receptor in specific cell types might be required.

Line 155. Provide an example of a bacterial species that produce histamine.

Line 202. CD11b+Ly6G+ cells are specific for mice. Is there a human counterpart?

Line 217-220. Provide reference to the sentence “Consequently, this hypothesis…”

Author Response

Dear editor,

Thank you very much for handling our submission and considering its publication in IJMS. We also sincerely would like to thank the reviewers for their thoughtful and constructive comments, which helped to substantially improve our manuscript. In the following, you will find a point-by-point explanation of the revisions we made according to the reviewers comments.

Reviewer 2:

In this review manuscript the authors have detailed functions of histamine, binding of its receptors, particularly H4R, with an emphasis on pathophysiological role in the gastrointestinal tract. It is a quite good summary of the current knowledge of the topic. The language could be slightly improved, and although already well references, there are sections without any. Find enclosed a list of a few point.

 Add a box with key points from the review.

Thanks for your kind evaluation and indicating the lack of references in some sections. We carefully revised the manuscript and introduced additional references where necessary.

Line 37. Mention other cells that produce histamine (Other cells, such as..)

Examples have been added.

Line 57 add “due to its expression pattern” to “Thus, [due to..] H4R is..”

The phrase has been added.

Line 88-91. Rephrase the sentences “At least chemical..” and “This requirement...” to improve the language.

The sentence has been rephrased.

Line 92: Is the H4R expression pattern different comparing humans and mice?

Such discussion has been added to the section H4R basics

On line 103 is detailed trials to block H4R. Are these trials for asthma and dermatitis as mentioned on line 101? In the abstract is mentioned trials for rheumatoid arthritis and psoriasis. Could the authors briefly add information of these.

We are sorry for these inaccuracies, which have been corrected in the revised version of the manuscript.

Line 107-108. Is H4R expressed on both human and mice eosinophils and mast cells?

Yes, it is. We added that notion at the indicated place.

Line 171. Related to genetic approaches to study H4R, particularly for studies in the gut (inflammation and carcinogenesis, eg. line 311-327): targeted approach to remove the receptor in specific cell types might be required.

Thanks for this suggestion. We added such discussion appropriately.

Line 155. Provide an example of a bacterial species that produce histamine.

Examples have been added.

Line 202. CD11b+Ly6G+ cells are specific for mice. Is there a human counterpart?

No, so far no human counterpart is known. We added such discussion at the corresponding place.

Line 217-220. Provide reference to the sentence “Consequently, this hypothesis…”

Corresponding references have been added.

Round 2

Reviewer 1 Report

All my concerns from my previous review have been addressed.